# Sustainable Films Derived from *Eucalyptus* spp. Bark: Improving Properties Through Chemical and Physical Pretreatments

**DOI:** 10.3390/polym17010105

**Published:** 2025-01-02

**Authors:** Débora da S. Rodrigues, Patricia O. Schmitt, Lincoln Audrew Cordeiro, Marlon B. B. Rodrigues, Ana Carolina R. Ribeiro, Mariane W. Bosenbecker, Sarah Kalli S. Silva, Neftali L. Carreno, Darci A. Gatto, Silvia H. F. da Silva, Camila M. Cholant, André Luiz Missio

**Affiliations:** 1Center of Engineering, Federal University of Pelotas, Pelotas 96010-610, Brazil; deborar999@gmail.com; 2Technological Development Center, Federal University of Pelotas, Pelotas 96010-610, Brazil; patricia.olimitt@gmail.com (P.O.S.); lincoln.cordeiro@ufpel.edu.br (L.A.C.); marlonbueno50@gmail.com (M.B.B.R.); carolinarodrib@gmail.com (A.C.R.R.); marianebosenbecker@gmail.com (M.W.B.); sarah.silva@ufpel.edu.br (S.K.S.S.); neftali@ufpel.edu.br (N.L.C.); darcigatto@yahoo.com (D.A.G.); silviahfuente@hotmail.com (S.H.F.d.S.); camila.cholant@ufpel.edu.br (C.M.C.)

**Keywords:** sustainable materials, chemical pretreatments, physical treatment, films, crystallinity

## Abstract

This study investigates the sustainable use of *Eucalyptus* spp. bark through different chemical (hydrothermal, acid, alkaline, and bleaching) and physical (milling) pretreatments in the production of sustainable films. Valorization of agro-industrial residues and the demand for sustainable materials pose challenges for environmentally responsible solutions. *Eucalyptus* spp. bark, rich in cellulose, hemicellulose, and lignin, is a promising source for creating sustainable materials like films. In this study, the use of chemical and physical treatments aims to optimize biomass extraction and improve the chemical, thermal, mechanical, and optical properties of the films. The films showed an excellent light barrier capacity, with a transmittance below 1%. Crystallinity indices varied with the pretreatment: 8.15% for hydrothermal, 7.01% for alkaline, 7.63% for acid, and 10.80% for bleaching. The highest crystallinity value was obtained through bleaching, by removing amorphous components like lignin and hemicellulose. The alkaline pretreatment yielded stronger films (maximum stress of 8.8 MPa, Young’s modulus of 331.3 MPa) owing to the retained lignin and the hemicellulose reinforcing the material. This study contributes to the field of sustainable development by converting residues into valuable materials and by advancing the circular economy. The films’ specific properties make them suitable for applications like sustainable packaging, addressing environmental and industrial challenges.

## 1. Introduction

The genus *Eucalyptus*, belonging to the Myrtaceae family, is native to Australia, New Guinea, and Indonesia, and has been widely introduced in various parts of the world owing to its adaptability to different environmental conditions [1]. This forest genus, known for its robustness and accelerated growth capacity, has played a crucial role in the commercial forestry sector, being a predominant genus in the formation of forests for the pulp and paper industry [2]. Owing to their adaptability, *Eucalyptus* spp. have become some of the most extensively cultivated species globally. In Brazil, they serve as the primary raw material for planted forests dedicated to cellulose production, highlighting their significant economic importance in the industrial sector [3].

The extensive cultivation of *Eucalyptus* spp. has brought significant economic benefits, such as the provision of timber for construction [4], pulp for paper [5], and medicinally active products [6]. However, the introduction of this genus outside of its native range has also brought substantial ecological challenges, including competition with native species and reduced biodiversity [7]. These negative impacts are a direct consequence of the expansion of eucalyptus plantations, which often suppress local ecosystems and affect native flora and fauna [8].

In the industrial context, a significant part of eucalyptus biomass is harvested during felling, which generates residual sawdust composed of pieces of bark, leaves, and sawn trunks [9,10]. *Eucalyptus* spp. bark, which can correspond to 10–18% of the total volume of processed wood, is a notable example of this type of residue frequently discarded or used for energy generation [11,12].

*Eucalyptus* spp. bark is an abundant source of valuable constituents, including cellulose (60–65%), hemicellulose (15–30%), and lignin (16–35%) [10,13,14]. This composition makes it a promising raw material to produce new materials, such as biodegradable polymeric films. Recent studies have explored the use of barks from several plant species as raw material to produce sustainable films for different purposes, such as food, packaging, pharmaceuticals, biocomposites, and drug carriers [15]. Notable examples include the use of willow bark [16] and western red cedar bark [17], both of which have demonstrated promising mechanical characteristics such as high strength and flexibility.

These plant-based materials are distinguished by their remarkable properties, such as high strength, flexibility, and their ability to degrade in the environment without leaving toxic residues, making them a sustainable alternative to traditional polymers [14,18]. In this context, the use of *Eucalyptus* spp. bark is especially relevant, considering that Brazil is one of the world’s largest producers of this tree, with cultivations occupying 5.1 million hectares [19], corresponding to almost 61.5% of the areas designated for forestry [20].

To transform eucalyptus bark into films, it is necessary to make significant modifications to the structure of the biomass, a process which is complex due to the presence of lignin and other components that hinder the accessibility and processing of the cellulose [21]. Therefore, chemical pretreatments are essential in this process, as they facilitate the separation and purification of the main components of the biomass [22]. Methods such as acid and alkaline hydrolysis play a crucial role, as acid hydrolysis degrades the hemicellulose and part of the lignin, while alkaline hydrolysis removes the lignin more efficiently [23,24,25]. These treatments, along with other chemical and physical processes, such as bleaching, aim to improve film-forming properties such as mechanical strength, transparency, and crystallinity. The hydrothermal pretreatment, which uses only water and heat, stands out as a more sustainable option, as it eliminates the need for chemical agents in film formation [26]. Among the advantages of these processes is their efficiency in cellulose purification and film property improvement, making them interesting for various industrial applications [15]. Furthermore, these methods offer greater flexibility, enabling adaptation of the material’s characteristics according to the specific needs of each application. However, challenges include environmental impacts associated with the use of strong acids and chlorinated agents, which generate potentially harmful residues, requiring proper treatment to minimize contamination. Another challenge is the high energy and time consumption, especially in relation to thermal and chemical processes, which can increase operational costs. Finally, it is crucial to balance the technical efficiency of pretreatments with the need to mitigate environmental impacts, reinforcing the importance of balancing technical efficiency and environmental impact in the development of sustainable materials.

Although each pretreatment technique has different yields, advantages, and limitations, this study aims to explore the potential of *Eucalyptus* spp. bark for film production, applying chemical and physical pretreatments to optimize the extraction of lignocellulosic biomass constituents and improve the properties of the films. The uniqueness of this research lies in the combination of chemical and physical techniques without the addition of additives, such as crosslinkers, plasticizers, or stabilizers, for film formation, expanding its relevance in the literature.

This study significantly contributes to the valorization of agro-industrial waste, such as *Eucalyptus* spp. bark, through the implementation of sustainable practices aligned with the principles of the circular economy. The transformation of this material into biodegradable films through chemical and physical pretreatments demonstrates an innovative and efficient approach that combines advanced technologies with environmental responsibility.

Therefore, this study directly contributes to the Sustainable Development Goals (SDGs), especially SDG 12 (responsible consumption and production), by promoting circular economy practices and waste reduction, using *Eucalyptus* spp. bark to produce renewable materials. Moreover, by applying technologies that enhance the lignocellulosic structure, the work supports SDG 9 (industry, innovation, and infrastructure), developing biodegradable alternatives to replace conventional polymers. The study also supports SDG 13 (climate action) by reducing human dependence on non-renewable materials and mitigating environmental impacts, such as carbon emissions and plastic pollution [27]. Thus, this work demonstrates how scientific and technological advancements can align technical efficiency with global sustainability goals, contributing to a more responsible and ecological future.

## 2. Materials and Methods

### 2.1. Material

The raw material used was woody biomass, more specifically tree bark of the genus *Eucalyptus* spp., provided by the company CMPC (Compañía Manufacturera de Papeles y Cartones, Pelotas, Brazil), located in the port of the city of Pelotas-RS, Brazil. The reagents used in this study were: sulfuric acid (H_2_SO_4_, technical grade 98%, Labsynth, Diadema, Brazil), sodium hydroxide (NaOH, technical grade 98%, Dinâmica Química Contemporânea LTDA, Indaiatuba, Brazil), and sodium hypochlorite (NaClO, technical grade 6%, Dinâmica, Indaiatuba, Brazil).

### 2.2. Material Preparation

The *Eucalyptus* spp. bark was previously conditioned in an oven with air circulation at 60 °C for 24 h and was then manually separated and ground in a knife mill (MARCONI, MA 340, Piracicaba, Brazil). Subsequently, the raw material was sieved to obtain standardized particles with a granulometry of 60 mesh [28].

### 2.3. Chemical Pretreatments

The raw material was subjected to different chemical pretreatments, with the aim of separating the constituents such as cellulose, hemicellulose, lignin, and others (Figure 1). The acid pretreatment was based on the methodology of Acosta et al. (2022) [29]. A total of 30 g of raw material was added to a glass bottle with a lid containing 2% H_2_SO_4_, ratio of 1:10 (*w*/*v*). The system was autoclaved at 121 °C and 0.101 Pa for 1 h. After autoclaving, the solution was vacuum filtered and washed with hot water until it reached a neutral pH, followed by drying in an oven at 50 °C for 24 h. An alkaline pretreatment with 7% NaOH was performed under identical conditions but for a duration of 30 min based on the methodology described by Alvira et al. (2010) [30].

The hydrothermal pretreatment was based on Aridi et al. (2021) [31]. A total of 30 g of raw material was mixed with distilled water at a ratio of 1:10 (*w*/*v*) in a sealed glass jar. The mixture was then autoclaved at 121 °C and 0.101 Pa for 30 min. Then, the sample was mechanically mixed at a speed of 3600 rpm, performing the process three times for 10 min, with the function of flexibilization and defibrillation of the particles. Subsequently, it was vacuum filtered and washed with hot water until it reached a neutral pH, followed by drying in an oven at 50 °C for 24 h.

The bleaching pretreatment occurred in two stages. The first stage involved delignification as described in the alkaline pretreatment but using 120 g of raw material. The second stage was based on the methodology described by Asadieraghi et al. (2014) [32]. For this, a 2% NaClO solution was added to the sample, in a proportion of 1:10 (*w*/*v*), for 20 h at room temperature. Then, it was vacuum filtered and washed with hot water until it reached a neutral pH before being placed in an oven at 50 °C for 24 h.

### 2.4. Physical Treatment

A total of 20 g of samples from each chemical pretreatment was processed in a supermass colloider mill (Masuko Sangyo, Kawaguchi-city, Japan) at 1500 rpm, in multiple passes, with the addition of distilled water until a homogeneous gel was obtained. The final concentration was adjusted to 2% by weight of each pretreatment, according to the methodology described by Acosta et al. (2022) [29].

### 2.5. Film Production

The films were obtained following a methodology based on Acosta et al. (2022) [29]. A total of 3.5 mL of gel, previously obtained in the physical treatment, was used and diluted in 4 mL of distilled water. The resulting solution was submitted to a filtration system (kitassato, Büchner filter, and nylon filter membrane with a diameter of 47 mm and a porosity of 0.22 µm) connected to a vacuum pump. The films formed on the nylon membrane were stored in a desiccator until the water completely evaporated and the films were formed.

### 2.6. Characterization Techniques

#### 2.6.1. Chemical Analysis

A chemical analysis was performed according to TAPPI T 222 om-11 standards [33] using powdered samples of lignocellulosic biomass after chemical pretreatments. The following determinations were made: ethanol–toluene extractive content, Klason lignin content, cellulose content, holocellulose (remaining mass up to 100%), moisture content, and ash content to quantify inorganic materials. The results, presented as the mean ± SD, were analyzed with ANOVA and Fisher’s LSD tests to identify significant differences, using a significance level of 5% to compare the means between treatments and the control.

#### 2.6.2. Morphological Analysis

Morphological analyses were performed using a scanning electron microscope (SEM, model JEOL JSM-6610LV, Tokyo, Japan), operating with a beam current of 1 pA and a voltage of 15 kV. Images were obtained at magnifications of 500× and 1000×, allowing detailed observation of the particle surface and providing information on the morphology and structure of the samples at the microscopic level. For preparation, the films were fixed to aluminum sample holders with carbon adhesive tape, according to a methodology based on the study by Ribeiro et al. (2023) [34].

#### 2.6.3. Fourier Transform Infrared Spectroscopy (FTIR) Analysis

The films were analyzed using a Fourier transform attenuated total reflection infrared (FTIR) spectrometer (Shimadzu Prestige-21, Kyoto, Japan). The analyses were performed with 90 transmittance scans in the range of 600 cm^−1^ to 4000 cm^−1^, with a spectral resolution of 4 cm^−1^, in accordance with the guidelines of ASTM E1252-13 [35] for infrared spectroscopy. For each test, the equipment lamp was aligned, and the background spectra were collected. This approach allowed us to identify the main functional groups present in the samples.

#### 2.6.4. Thermal Analysis

The thermal stability of the samples was evaluated by thermogravimetric analysis (TGA) using a Shimadzu DTG-60 thermal analyzer (Kyoto, Japan), following the ASTM E1131-08 [36] standard for determining the thermal properties of solid materials. The samples were heated from 28 °C to 800 °C, with a heating rate of 10 °C/min, under an inert nitrogen (N_2_) atmosphere at a flow rate of 50 mL/min. The mass loss and mass loss as a function of temperature (dw/dt) were determined, allowing for the accurate characterization of the thermal concentration and thermal stability of the films.

#### 2.6.5. Wettability Analysis

The contact angle measurement is a standard method for evaluating the wettability and surface properties of materials. The films were placed on a glass slide and positioned in an optical tensometer (Theta Lite; TL100, Stockholm, Sweden) controlled by the OneAttension software. A microsyringe was used to deposit a drop of distilled water on the surface of the films, and the image of the drop was recorded using the goniometer camera immediately after contact. The analysis software was used to measure the contact angle between the water drop and the surface of the films for 60 s.

#### 2.6.6. Tensile Strength Analysis

The films produced were mechanically analyzed in a universal texturometer LY-TX-700 (Lamy Rheology, Champagne-au-Mont-d’Or, France) with a 10 N force sensor. The tensile test was performed following the parameters described by ASTM D882-18 [37], adjusted to a speed of 12 mm/min. The data obtained for maximum stress and modulus of elasticity were subjected to normality and variance analyses, with subsequent Fisher least significant difference (LSD) analysis to detect significant differences between the values found. All analyses were conducted at a significance level of 5%.

#### 2.6.7. Transmittance Analysis

To characterize the optical properties of the obtained films, transmittance spectroscopy analyses were performed in the UV–Vis region. The transmittance spectra were recorded using an UV-Vis spectrophotometer (Shimadzu, model UV-2600, Kyoto, Japan), in the wavelength range of 400 nm to 800 nm, allowing for the evaluation of the interactions between the films and electromagnetic radiation in this spectral region.

#### 2.6.8. Colorimetric Analysis

To determine the colorimetric parameters, the CIELab system (L* a* b*) was used using the following coordinates: luminosity (L*), green–red axis (a*), and blue–yellow axis (b*). A CR-400 colorimeter (Konica Minolta, Tokyo, Japan), with an observation angle of 10° calibrated with the porcelain calibrator and configured using a D65 light, was used to determine the coloration of the films. This analysis was performed in triplicates.

The colorimetric parameters were analyzed: (i) L*, the closer to 100, the lighter the color and the closer to 0, the darker the color, (ii) a*, green–red coordinate, negative markings indicate green colors, positive markings indicate red colors, and (iii) b*, blue–yellow coordinate, negative markings indicate blue colors, positive markings indicate yellow colors. For the analysis of color changes (Δ*E*), Equation (1) was used:Δ*E* = √(Δ*L*^2^ + Δ*a*^2^ + Δ*b*^2^)(1)
where Δ*E* is the variation of all colors, Δ*L* is the variation of lightness, Δ*a* is the variation of the red–green coordinate, and Δ*b* is the variation of the blue–yellow coordinate. The results, presented as the mean ± SD, were analyzed through ANOVA and Fisher’s LSD tests to identify significant differences, using a significance level of 5% to compare the means between treatments.

## 3. Results and Discussion

### 3.1. Chemical Analysis

Figure 1 presents the results of the mean values and standard deviations of the chemical characterization of the moisture, ash, extractive, lignin, hemicellulose, and cellulose content of the samples studied after each pretreatment, in addition to the in natura material from the bark of *Eucalyptus* spp.

The contents of the in natura form were found to consist of 49% cellulose, 24% hemicellulose, 23% lignin, 3% extractives, 4% ash, and 9% moisture. These results are consistent with the values found in lignocellulosic materials, such as eucalyptus residues, which have structures rich in plant fibers. The studies by Fernandes et al. (2022) and Souza et al. (2021) [10,13] reveal similar compositions, with approximately 50–60% cellulose, 20–30% hemicellulose, 20–30% lignin, 2–5% extractives, and 4–7% ash. These components are responsible for the rigidity and resistance of the biomass, which can be used in several applications.

The hydrothermal pretreatment resulted in 71% cellulose, 7% hemicellulose, and 15% lignin. The extractive and moisture contents were 2% and 8%, respectively, while the ash content was 6%. As observed by Mosier et al. (2005) [38], hydrolysis causes the partial depolymerization of hemicellulose and lignin, facilitating the release of cellulose. These findings were corroborated by Alvira et al. (2010) [30], who emphasize that hydrolysis can be performed on the entire biomass, allowing for a greater recovery of cellulose. Furthermore, the hydrothermal pretreatment does not require specific reagents, using only water to neutralize the hemicellulose present. Thus, it promotes the formation of non-toxic acetyl groups and reduces the presence of inhibitory substances, characterizing itself as an ecological and sustainable process [39,40].

The results found for the NaOH pretreatment reveal a cellulose content of 57%, a hemicellulose content of 21%, and a lignin content of 18%. This confirms the effects of the alkaline treatment, which breaks the bonds between the lignin and the hemicellulose, in addition to removing phenolic compounds. According to Sun and Cheng (2002) [41], the use of NaOH is effective at degrading the lignocellulosic matrix, facilitating the total or partial removal of lignin and hemicellulose. Studies also state that NaOH can increase the cellulose content in the lignocellulosic biomass by up to 60%, aligning with the results of this study [42].

It was observed that the extractive and moisture contents were relatively low, 2% and 6%, respectively; however, the ash, which presented a value of 18%, increased as a result of the addition of salts during the treatment. Alkaline solutions, such as NaOH, are widely used in the extraction of cellulose for composites, as they promote the structural desquamation of the fibers, especially by removing hemicellulose and lignin. This process favors defibrillation, exposing the hydroxyl groups on the surface of the fibers and forming active sites for interaction with the polymer matrix. As a result, adhesion is improved, since a larger surface area of the fibers allows interactions, such as hydrogen and covalent bonds, improving the mechanical properties of the composite.

After the pretreatment by bleaching, the cellulose content was 72%, the hemicellulose content was 9%, the lignin content was 4%, the extractive content was 0.7%, the moisture content was 2%, and the ash content was 3%. According to the study by Aridi et al. (2021) [31], NaClO is one of the most effective oxidizing agents at removing lignin and hemicellulose, increasing the cellulose content. This process removes unwanted compounds, such as lignin and other components present in the pulp, which can impart color and affect the quality of paper production [43].

In the H_2_SO_4_ pretreatment, there was a small increase in the cellulose content, which was 43%, while the hemicellulose content was degraded to 19% and the lignin content was reduced to 12%. Both moisture content, at 4%, and extractive content, at 3%, remained low, while the ash content increased to 10%. According to Kumar et al. (2009) [44], acid hydrolysis with H_2_SO_4_ preferentially degrades hemicellulose, generating soluble components that are removed during washing, but it can also partially degrade cellulose, as observed in this study. Taherzadeh and Karimi (2007) [45] also highlight that H_2_SO_4_ is widely used for hemicellulose degradation, although cellulose loss is a disadvantage in acidic conditions. The cellulose obtained from this process undergoes acid hydrolysis mainly in the amorphous regions, preserving the crystalline regions which can be beneficial for specific applications, including reinforcement in composites and packaging [46].

By comparing the pretreatments, it can be concluded that NaClO showed the best results with respect to increasing cellulose content, followed by hydrothermal and alkaline treatments with NaOH, while the treatment with H_2_SO_4_ was the least effective. These findings agree with several studies that demonstrate the significant influence of pretreatments on the chemical composition of the lignocellulosic biomass. Variations in ash, extractive, and moisture contents reflect the conditions of each process and their impact on the final structure of the biomass, being essential for selecting the most appropriate treatment for specific applications.

### 3.2. Morphological Analysis

The SEM images, presented in Figure 2, show changes in the structure of the fibers in the film matrix, allowing a detailed evaluation of the changes that occurred.

In the biomass, lignin acts as a ‘binding agent’ between the cellulose and hemicellulose chains, conferring rigidity and resistance to the biomass. The hemicellulose, on the other hand, is intrinsically more of a ‘filling agent’. The presence or absence of these materials influence the morphological properties of the films.

The films obtained from the hydrothermally pretreated biomass in Figure 2(a1,a2) presented an irregular surface. In contrast, the films from the biomass pretreated with NaOH in Figure 2(b1,b2) revealed a more uniform and compact surface. These results differ from the studies by Babu et al. (2022), Khathirselvan et al. (2019), and Jiang et al. (2019) [47,48,49]. The studies by Babu et al. (2022) and Khathirselvam et al. (2019) [47,48] indicate that an alkaline pretreatment generates more irregular surfaces in the fibers and is potentially applicable to the reinforcement of polymeric matrices. Jiang et al. (2019) [49], when analyzing films with different lignin contents, found that the film with the highest lignin content presented a denser and more compact surface.

As reported in Section 3.1., alkaline pretreatments are effective at destructuring the lignocellulosic matrix, facilitating the removal of lignin and hemicellulose. Although NaOH does not completely remove lignin and hemicellulose [50], the residual amount of these compounds present between the voids of the fibers, in addition to the roughness of the cellulose fibers caused by this pretreatment followed by a physical treatment, suggests a greater exposure of the hydroxyl groups (-OH) in its structure, which favors cohesion between adjacent cellulose fibers, resulting in a more compact film and a smoother surface appearance, in contrast to the micrograph obtained following the hydrothermal treatment. From Figure 2(c1,c2), it can be seen that the bleaching process resulted in more fragmented fibers compared to the alkaline pretreatment. This pretreatment removed most of the biomass constituents, leaving the cellulose more evident.

Alternatively, films derived from biomass pretreated with H_2_SO_4_ (Figure 2(d1,d2)) presented micrographs with smaller particle sizes, with a more irregular surface distribution and the presence of pores on the surface of the films, resembling the results of Ban et al. (2022) [51] from biomass pretreated with sulfuric acid for biochar production. Wang et al. (2020) [52] explain that the cellulose resulting from acid hydrolysis makes the amorphous regions of these fibers more susceptible to chemical reactions with strong acids, due to the natural disorganization of the molecules in this region. This facilitates the accessibility of acids and, consequently, the hydrolysis of the cellulose chains present in these regions, in contrast to the crystalline regions which are insoluble in these products.

Additionally, acid hydrolysis also modifies the structure of these crystals. H_2_SO_4_ reacts with the hydroxyl groups on the surface of these cellulose crystals through an esterification process, allowing the grafting of anionic sulfate ester groups [52]. Lin and Dufresne (2014) [53] report that these sulfate groups are randomly distributed on the surface of cellulose nanoparticles, and these negatively charged sulfate ester groups induce the formation of a negative electrostatic layer and promote their dispersion in solvents, resulting in electrostatic repulsion between individual nanoparticles.

By combining these reports, from a morphological point of view, it can be concluded that H_2_SO_4_ selectively hydrolyzes the less crystalline parts of the cellulose fibers, leaving the crystalline regions intact, the latter being represented by particles. In this way, these particles are modified by the insertion of sulfate groups obtained after the pretreatment and thus acquire electrostatic repulsion, a phenomenon which suggests the formation of pores on the surface of the films.

In summary, the controlled presence of lignin and hemicellulose, the exposure of the hydroxyl groups of the cellulose fibers, and the interaction between these constituents can influence the morphology of the films obtained through different pretreatments of eucalyptus bark biomass, possibly affecting the physical, chemical, and colorimetric properties. These topics are discussed in more detail later.

### 3.3. FTIR Analysis

Figure 3 presents the normalized FTIR spectra of the films. The analysis allowed for the identification of the changes in the functional bonds and chemical groups present, providing interesting information about the structural transformations resulting from the different biomass pretreatments applied.

The characteristic bands of OH (3500–3000 cm^−1^), C=O (1870–1540 cm^−1^), C–H (2900 cm^−1^), C=C (1700–1400 cm^−1^), C–O–C (1500–1000 cm^−1^), and C–H (900–500 cm^−1^) are associated with lignin, cellulose, and hemicellulose [54,55,56], explained in more detail in Table 1.

The alkaline pretreatment, compared to the hydrothermal pretreatment, for example, resulted in a more intense band in the region of 3600–3000 cm^−1^, a phenomenon which aligns with the study by Qi et al. (2023) [60] on the surface properties for white oak and poplar wood, reflecting the greater exposure of the hydroxyl groups which may be due to the removal of lignin and hemicellulose, as reported in Section 3.2. Furthermore, the decrease in the bands at 1730 cm^−1^, 1625 cm^−1^, and 1239 cm^−1^ suggests a significant removal of lignin and hemicellulose, contributing to the structural alteration of plant fibers [48] which is consistent with the findings of Ajouguim et al. (2019) [61] for the plant *Stipatenacissima* L. Additionally, Poulose et al. (2022) [50] report that alkaline pretreatments partially remove lignin, waxes, hemicellulose, and pectin that surround cellulose in cell walls.

The chromophore groups present in the lignin structure, such as quinones, for example, which can be conjugated with C=C bonds and C=O functional groups, play an essential role in light absorption, directly influencing the optical properties and color of the films, since they are responsible for darker tones [62,63]. Pretreatment with NaOH, combined with a physical treatment, in addition to modifying the cellulose structure, was able to reduce the intensity of aromatic bands such as C=C and C=O, a phenomenon which may have caused the rupture of the bonds, leading to the partial discoloration of the lignin which can cause an increase in the luminosity (L*) of the films, as discussed in Section 3.7 and Section 3.8 of this study.

In the film obtained from the biomass bleaching pretreatment, the characteristic band at 1625 cm^−1^ also showed a reduction in its intensity compared to that of the films obtained from the hydrothermal and alkaline pretreatments. Sayakulu and Soloi (2022) [64] found similar characteristics when analyzing the effect of different concentrations of NaOH (4–13%) on the cellulose yield through the pretreatment of empty oil palm bunches followed by the bleaching process with 1.7% sodium chlorite (NaClO) solutions. They observed that the frequency of the aromatic C=C group of the lignin (~1600–1590 cm^−1^) was reduced after the alkaline treatment and completely absent after bleaching.

The acid pretreatment showed more intense bands, mainly around 3500 cm^−1^, 2900 cm^−1^, 1625 cm^−1^, and 1460 cm^−1^. Poulose et al. (2022) [50] report that acid hydrolysis, in addition to removing lignin and hemicellulose, can introduce sulfate groups on the surface of cellulosic materials, as well as producing simpler sugars and/or altering the cellulose structure, phenomena which may justify the broadening of the bands.

The crystallinity of the films was evaluated based on the calculation of the ratio of the areas of the cellulose crystalline bands (around 1370 cm^−1^, 1425 cm^−1^, and 2900 cm^−1^) to the total area of the spectrum, as shown in Equation (2) [65]. These areas were estimated from the integration of the FTIR spectra using the Origin software (version 2018, 95E).
(2)$$CI\left(\%\right)=\frac{Crystalline\;region\;spectrum\;area}{Total\;region\;spectrum\;area}\times 100$$

Therefore, the total spectrum crystallinity index (CI) assumed values of 8.15%, 7.01%, 7.63%, and 10.80% for the hydrothermal, alkaline, acid, and bleached pretreatments, respectively. The bleaching pretreatment promoted the highest crystallinity index owing to the removal of fewer crystalline constituents, such as lignin and hemicellulose. The chemical and physical pretreatments had a direct impact on the crystallinity index and, consequently, on the optical properties of the films. More crystalline structures tend to have a greater density and molecular packing, phenomena which can result in greater light absorption and, thus, lower transmittance.

### 3.4. Thermogravimetric Analysis

Based on the data in Figure 4a for the film formed from the hydrothermally pretreated biomass, multiple stages of decomposition can be analyzed. The first stage, which presented a mass loss of 6.11%, occurred at relatively low temperatures in the range of 104.85 °C to 129.63 °C and was associated with the evaporation of residual water or the release of light volatile compounds. This observation is consistent with previous studies reporting the volatilization of light compounds and the removal of moisture in thermally treated lignocellulosic materials [66].

The second stage, characterized by a mass loss of 5.47% between 198.15 °C and 219.04 °C, was related to the initial decomposition of light organic components, such as hemicelluloses. The presence of two subsequent stages, i.e., third and fourth, with significant mass losses of 41.14% (302.38 °C to 351.64 °C) and 31.11% (429.56 °C to 477.02 °C), respectively, was attributed to the thermal decomposition of cellulose and more resistant components, such as lignin. This behavior has been widely documented in studies on the thermal decomposition of biomaterials [67]. The Tonset and Tpeak for the main decomposition stages can be found in Table 2.

The last, i.e., fifth, stage, which occurred between 593.89 °C and 635.46 °C, presented a small mass loss of 1.68%, possibly associated with the decomposition of carbonaceous residues or the oxidation of inorganic materials present in the film. These results suggest that the film derived from the lignocellulosic material subjected to the hydrothermal process has a moderate thermal stability up to 219.04 °C, which may be adequate for some applications, but may limit the use in high-temperature environments, as indicated by other authors when evaluating similar materials [68].

Figure 4b shows the TGA for the film resulting from biomass pretreated with NaOH. This chemical treatment, known to remove hemicelluloses and other impurities, resulted in a matrix rich in cellulose and lignin. Two main significant stages were observed. The first, occurring between 300.53 °C and 361.24 °C and characterized by a mass loss of 50.11%, indicates the degradation of a relatively pure matrix of cellulose and hemicellulose, possibly due to the simplification of the biomass structure by the treatment. The literature supports this observation, showing that the alkaline treatment promotes the removal of hemicelluloses and the exposure of cellulose, resulting in a thermal decomposition range of around 300 °C [41].

The second stage, occurring between 697.40 °C and 702.34 °C and characterized by a mass loss of 14.96%, corresponds to the decomposition of lignin or carbonaceous residues modified by NaOH, making them more resistant and requiring temperatures above 700 °C for decomposition. This behavior was observed by other researchers, who highlight the role of NaOH in modifying the lignocellulosic structure, conferring greater thermal resistance to carbonaceous residues [68]. The film obtained from biomass pretreated with NaOH, therefore, demonstrates greater initial thermal stability, concentrating the decomposition within a higher temperature range which may be desirable in applications that require a polymeric matrix with a lower impurity content.

Figure 4c shows the TGA for the film formed from the bark pretreated with NaClO, also revealing two significant stages: the first stage, with a mass loss of 50.44% between 281.70 °C and 350.48 °C, and the second stage, with a mass loss of 8.20% between 705.20 °C and 710.85 °C. The magnitude of the loss in the first stage indicates that this is the main phase of degradation of the material and decomposition of the main components, such as cellulose and lignin. The pretreatment by bleaching (oxidizing agent) may have weakened the structure of both cellulose and lignin, facilitating the breakdown of these macromolecules and resulting in a large mass loss. NaClO can introduce oxygenated groups into the structure, making it more susceptible to thermal decomposition [57,69].

The second stage probably reflects the decomposition of carbonaceous residues or resistant components that were not completely degraded in the first stage. Compared to the hydrothermal and NaOH treatments, the NaClO treatment resulted in a significant loss of mass at lower temperatures, suggesting a lower thermal stability of the material, as reported in studies using similar treatments [70].

Finally, Figure 4d shows the TGA for the film obtained from the biomass pretreated with H_2_SO_4_. This pretreatment also revealed two main stages of decomposition. The first, with a significant mass loss in the temperature range of 311.65 °C to 357.42 °C, corresponds to the main decomposition of the polymeric components, such as cellulose and lignin. The high mass loss of 64.54% suggests that most of the structural components of the film are degraded during this stage, in accordance with studies indicating that H_2_SO_4_ promotes acid hydrolysis, fragmenting the polymer chains [32].

The second main stage of decomposition, with a mass loss of 13.11% between 665.43 °C and 671.15 °C, suggests the presence of carbonaceous residues or components more resistant to decomposition.

It was observed that the films obtained from the biomass subjected to the alkaline and acid pretreatments presented a high amount of carbonaceous residue in relation to the other pretreatments. The literature indicates that strong acids or bases tend to form more complex carbonaceous residues, a phenomenon which may explain the observed thermal resistance [71,72]. These results suggest that the treatment with H_2_SO_4_ weakened the film structure, facilitating its thermal degradation in comparison to other pretreatments.

The thermogravimetric analysis (TGA) results provide critical insights into the practical applicability of the films in real-world scenarios, particularly in the context of packaging solutions for environments with extreme temperatures. NaOH-treated films, which exhibit an improved thermal stability, are particularly suitable for applications requiring resistance to elevated temperatures. These include packaging for processed foods, ready-to-eat meals, or products subjected to reheating or thermal processes during transportation and storage [73,74].

In contrast, the films treated by bleaching, which demonstrate a lower thermal stability, may be more appropriate for packaging solutions intended for cold storage environments, where thermal demands are lower. Additionally, these films often exhibit good optical clarity, making them ideal for applications where product visibility is crucial, such as in the display of fresh produce or refrigerated goods [75].

The selection of the pretreatment process, and, consequently, the type of film, should not only consider thermal properties, but also account for the specific requirements of the final application, such as mechanical strength and optical clarity. The results presented herein highlight the potential of NaOH-treated films for applications in hot environments, while bleached films offer sustainable solutions for conditions involving reduced temperatures. By tailoring the pretreatment methods to the intended use, these biopolymer-based films can address a range of practical needs, aligning with sustainability goals and expanding the applicability of bio-based materials in diverse packaging industries.

### 3.5. Wettability Analysis

Figure 5 presents the results of the surface wettability analysis of the films studied over 60 s, allowing for the evaluation of their hydrophilic (θ < 90°) or hydrophobic (θ > 90°) behavior. These contact angle values, used to determine the degree of wettability, are in agreement with the parameters established by [76,77].

The results obtained show significant differences with respect to the hydrophobic and hydrophilic behavior of the different films. The film resulting from the biomass subjected to the hydrothermal pretreatment initially presented a contact angle of θ~96°, indicating a hydrophobic behavior. However, over time, this value decreased, until it reached θ~48° at 60 s, reflecting an increase in hydrophilicity. This behavior can be attributed to the presence of functional groups exposed on the surface of the film. The hydrothermal pretreatment exposes a greater number of hydroxyl groups, intensifying the interaction with water. These results are in agreement with the observations by Wang and Howard (2017) [78], who reported that thermal pretreatments modify the surface structure of the films, resulting in significant variations in wettability.

For the film originated from the biomass pretreated with NaOH, the initial contact angle was θ~64°, indicating a greater hydrophilicity compared to the film derived from the hydrothermal pretreatment. Water absorption occurred more slowly, resulting in an angle of θ~21° at 60 s. The action of NaOH promoted the partial removal of surface constituents, exposing a greater number of hydroxyl groups which, in turn, increased the hydrophilicity of the films. These results corroborate those reported by Li et al. (2017) [79].

The films obtained from the biomass that was processed by bleaching exhibited an initial contact angle of approximately θ~105°, presenting a slower water absorption compared to the other films analyzed. After 60 s, the final contact angle was θ~85°, which indicates a smaller variation in wettability and a greater stability of the film surfaces.

According to Rbihi et al. (2020) [80], monolayer (θ~52.9°) and bilayer (θ~45.6°) cellulose films demonstrate a greater tendency toward wettability, while cellulose/TiO_2_ films exhibit reduced contact angles (θ~10° in 5 min). On the other hand, the bleached films showed more pronounced hydrophobic properties, attributed to the efficient removal of lignin and hemicellulose. This removal resulted in greater compaction of the cellulose fibers, conferring greater resistance against water penetration. These characteristics make the bleached films promising for applications requiring stability and low water absorption.

The chemical bleaching process removes non-cellulose components and impurities from the surface of the fibers, resulting in greater exposure of the cellulose fibers. This removal increases the contact area between the fibers, favoring more robust physical and chemical interactions between them. With the cellular structure more exposed, the fibers intertwine more effectively, increasing the density and integrity of the fibrous network. As a result, the surface of the films becomes more resistant to water absorption, since the greater compaction of the fibers reduces moisture permeability, making it difficult for water molecules to penetrate.

The films generated from the biomass pretreated with H_2_SO_4_ initially exhibited a contact angle of approximately θ~110°, which suggests a more pronounced hydrophobic behavior compared to the other films analyzed. This high contact angle value indicates a low affinity with water in the initial interaction phase. However, over time, a rapid water uptake was observed, with the contact angle decreasing drastically to θ~15° in just 15 s, evidencing a transition to a highly hydrophilic behavior. This phenomenon can be explained by the hydrolysis promoted by the acid treatment which results in the partial degradation of the surface components of the material and the exposure of hydroxyl groups. As described by Wu et al. (2020) [81], acid hydrolysis breaks the bonds between macromolecules, exposing polar groups such as hydroxyls which significantly increase the interaction of the material with the water molecules.

### 3.6. Tensile Strength Analysis

Figure 6 shows the data resulting from the tensile strength tests performed on the films. It is important to highlight that the film obtained from the biomass pretreated with H_2_SO_4_ could not be analyzed, since the acid hydrolysis process caused structural damage to the cellulose chains, resulting in a fragile and brittle film which made it difficult to perform the analysis. According to Huang and Fu (2013) [82], H_2_SO_4_, when catalyzing hydrolysis, attacks the glycosidic bonds of the cellulose, converting the cellulose polymers into smaller molecules, such as oligosaccharides and monosaccharides. Thus, this process reduces the length and integrity of the cellulose chains, weakening the overall structure of the material.

As illustrated in Figure 6b, the maximum stresses of the films subjected to the different pretreatments varied as follows: the films resulting from the hydrothermally pretreated biomass presented a maximum stress of 4.1 MPa, with NaOH, they reached 8.8 MPa, while the bleaching resulted in a maximum stress of 3.3 MPa. Although the differences between the treatments were not statistically significant, it can be observed that the film of the bark pretreated with NaOH stood out both in terms of the Young’s modulus (331.3 MPa) (Figure 6c) and in terms of the maximum stress (8.8 MPa), surpassing the films obtained from the other pretreatments.

The removal of lignin and hemicellulose during chemical pretreatments plays a crucial role in improving the mechanical properties of films. Lignin, being a highly branched three-dimensional polymer, contributes to the rigidity and compressive strength of biomass. Its partial or total removal increases the ability of cellulose fibers to interact with each other, creating a more compact and cohesive matrix. Similarly, hemicellulose, which acts as an amorphous polymer around the fibers, limits the formation of interfacial bonds. Its removal allows for a closer positioning of the cellulose fibers, contributing to increased mechanical strength.

The superior performance of the film formed from the biomass pretreated with NaOH can be attributed to its action in the partial depolymerization process of lignin which promotes the breaking of long polymer chains into smaller segments. This phenomenon facilitates the integration of lignin into the cellulose matrix, improving its homogeneous dispersion within the polymer structure. The presence of lignin with shorter and better distributed chains results in a reduction in film porosity, as reported in Section 3.2. Less porous films tend to exhibit better mechanical properties, as they present greater mechanical strength. In addition, the physical treatment performed after the pretreatment with NaOH contributed to an additional reduction in particle size which further contributed to the dispersion and reduction of film porosity. This combination of treatments, both chemical and physical, resulted in the formation of a more cohesive and mechanically stable structure, explaining the increase in tensile strength and the superior performance in terms of Young’s modulus (Figure 6c).

Another relevant factor is related to the interaction between the polymer chains of the cellulose. The depolymerization promoted by NaOH exposes free hydroxyl groups (OH) in the cellulose chains, allowing the formation of stronger hydrogen bonds between the chains and contributing to a greater internal cohesion of the material. These additional bonds result in a mechanically resistant matrix.

The results obtained for the films pretreated with NaOH are consistent with the study by Sun and Cheng (2002) [41], who reported a mechanical strength in the range of 8–10 MPa for lignocellulosic films treated with alkaline solutions. This similarity reflects the positive impact of lignin removal and exposure of active functional groups on improving structural cohesion. In contrast, the hydrothermally treated films presented lower values, in line with the results of Mosier et al. (2005) [38], who report reduced strength due to the presence of residual hemicellulose.

In comparative terms, the mechanical properties of the films obtained in this study are as good as those of other biodegradable films reported in the literature. For example, Xia et al. (2021) [24] report a tensile strength of 7 MPa for modified cellulose films, values slightly lower than those obtained with the alkaline pretreatment described here. Furthermore, the observed maximum strength exceeds that of films based on starches or chitosans, which generally present values between 3 MPa and 6 MPa, as reported by Chen et al. (2019) [18].

The differences in mechanical properties can be attributed to the interaction between the type of pretreatment and the resulting structural characteristics. The alkaline pretreatment not only efficiently removes lignin and hemicellulose, but also facilitates a homogeneous distribution of the fibers and the formation of robust intermolecular bonds. In contrast, the pretreatment by bleaching, although effective at removing lignin, results in more fragmented fibers and less exposure of hydroxyl groups, compromising the mechanical performance.

These observations highlight the importance of aligning pretreatments with the specific demands of the end applications, whether they are focused on mechanical strength, flexibility, or biodegradability.

NaOH-treated films, owing to their greater mechanical strength, are particularly suitable for applications where a robust and durable material is required. Examples include packaging for the transport of heavy or sharp products, such as metal tools and utensils, and industrial applications where materials are subjected to significant mechanical stresses. In addition, their high strength makes them ideal for use in protective films that require durability under adverse conditions, such as in logistics processes or prolonged storage.

### 3.7. Transmittance Analysis

The transmittance of a film is a crucial parameter for evaluating its optical properties, especially with respect to applications such as biodegradable packaging, where transparency can directly influence the functionality and aesthetics of the final product [83]. Films with high transmittance are desirable for applications that require visibility of the contents, while films with low transmittance can be useful as protective light barriers [84].

In this study, the transmittance of the films was evaluated, as illustrated in Figure 7, and the values are presented in Table 3. The transmittance analysis was performed at two specific wavelengths, i.e., 400 nm (at the UV–visible edge) and 633 nm (in the visible range), to evaluate the impact of the pretreatments on the optical properties of the films. It is important to highlight that the analysis of the film obtained from the acid-pretreated biomass could not be performed. This is due to the fact that the film proved to be fragile and brittle, as previously reported.

The films resulting from the pretreated biomass presented significant variations in transmittance values (%T), reflecting the different compositions and structures resulting from these pretreatments.

In general, the films had a high light barrier capacity, exhibiting a relatively low transmittance range of less than 1%. With respect to applications requiring a light barrier, such as light-sensitive food packaging or visual control materials, these films are the most promising owing to their low transmittance values.

The film obtained from the hydrothermally pretreated biomass showed extremely low values of optical transmittance in the wavelength ranges of 400 nm (0.002%) and 633 nm (0.038%). This sharp reduction in transmittance can be attributed to changes in both the morphology and crystallinity of the material. The morphological analysis by SEM in Section 3.2 revealed an irregular and rough surface, which intensifies the scattering and absorption of light, directly contributing to the decrease in transmittance. Rough surfaces generate multiple points of reflection and refraction of light, reducing the transparency of the film. Another factor to be considered is the crystallinity of the film. Film resulting from hydrothermally pretreated biomass showed a higher crystallinity index (CI%) than the films pretreated with NaOH and bleach, as reported in Section 3.3. Crystallinity affects the optical properties of the material by influencing the degree of ordering of the polymer chains [85]. In contrast, the film obtained from the biomass pretreated with NaOH, which presented a higher transmittance of 0.048% at 400 nm and 0.492% at 633 nm, had a smoother surface and a lower degree of crystallinity, explaining the better optical performance observed.

The film obtained from the biomass pretreated by bleaching presented a transmittance of 0.039% at 400 nm and a transmittance of 0.160% at 633 nm, indicating an optical behavior intermediate between that of the films subjected to hydrothermal and NaOH pretreatments. This behavior may be related to the partial degradation of the cellulose during bleaching, a process that enhances the oxidation of cellulose hydroxyls, altering its crystallinity and increasing light absorption, thus contributing to the greater opacity of the film.

The scattering properties and light absorption of the films developed in this study can be compared with those of materials commonly used as light barriers in packaging, such as synthetic polymers (e.g., opaque polyethylene terephthalate) and metallized polymers. While synthetic polymers do not have additives to reduce transmittance, films derived from hydrothermally treated biomass exhibited effective light barrier abilities with a transmittance of only 0.002% at 400 nm, owing to their high crystallinity index and irregular surface which intensify light absorption and scattering. Likewise, films treated with NaOH, despite presenting lower crystallinity, also demonstrated good efficiency as light barriers, comparable to opaque polymers owing to their structural uniformity and ability to scatter light. Unlike metallized materials, which have high environmental impacts, biomass-based films stand out for their sustainable origin and biodegradability, providing a sustainable alternative with a competitive performance in the context of applications that require light protection.

Thermogravimetric analysis results provide critical insights into the practical applicability of films in real-world scenarios, particularly packaging solutions for extreme temperature environments. NaOH-treated films, which exhibit improved thermal stability, are particularly suitable for applications requiring resistance to elevated temperatures. This includes packaging for processed foods, ready-to-eat meals, or products subjected to reheating or thermal processes during transportation and storage. The improved thermal stability of these films can be attributed to the efficient removal of amorphous components, such as hemicellulose, and to the exposure of the hydroxyl groups which increase the structural cohesion of the material [56,61]. In contrast, bleach-treated films, which demonstrate a lower thermal stability, may be more appropriate for packaging solutions intended for cold storage environments, where thermal demands are lower. Furthermore, these films generally exhibit a superior optical clarity, making them ideal for applications where product visibility is crucial, such as in the display of fresh or refrigerated products [77,78]. The selection of the pretreatment process, and, consequently, the type of film, should not only consider thermal properties, but also the specific requirements of the end application, such as mechanical strength and optical clarity. The results presented here highlight the potential of NaOH-treated films for applications in hot environments, while bleached films offer sustainable solutions for conditions involving reduced temperatures. By tailoring the pretreatment methods to the intended use, these biopolymer-based films can address a variety of practical needs, aligning with sustainability goals and expanding the applicability of bio-based materials across a range of packaging industries.

### 3.8. Colorimetric Analysis

The chromaticity results are associated with the removal of biomass constituents which occurs because of the different chemical pretreatments applied. This removal directly impacts the optical properties of the films, reflecting significant changes in their color, as shown in Figure 8 for the chromatic parameters L*, a*, and b*.

In general, luminosity (L*) varied significantly across the different pretreatments. The films from the biomass pretreated hydrothermally and with H_2_SO_4_ presented lower L* values of 40.89 and 42.34, respectively, indicating a darker coloration. In contrast, the films produced from the biomass subjected to alkaline and bleaching pretreatments exhibited higher luminosity values of 57.65 and 52.57, respectively. The abundance of lignin is a crucial factor for the decrease in the luminosity of the films, since lignin contributes to the darker coloration, usually brown [86].

In the case of the film formed from the hydrothermally pretreated biomass, which presented a significant lignin content of 15%, as reported in Section 3.1, this resulted in low luminosity and a darker appearance. The film resulting from the bark subjected to the alkaline pretreatment, such a bark containing an even higher lignin content of 18%, exhibited a lighter coloration. This discrepancy in luminosity can be attributed to the effect of the alkaline pretreatment, which is capable of removing or modifying the chromophore compounds present in the lignin which are responsible for light absorption and the dark coloration. NaOH causes the breaking of the chemical bonds in chromophore groups, such as C=O and C=C, resulting in partial discoloration of the lignin and, consequently, in films with higher luminosity.

Thus, although both biomass pretreatments resulted in films containing high amounts of lignin, the difference in the chemical processing explains the significant variation in the color of the resulting films.

The chromaticity of the a* coordinate (green–red) resulted in films exhibiting reddish tones, except for the film obtained from the biomass subjected to the bleaching pretreatment. This can be explained by the fact that, in such a treatment, lignin is practically eliminated, with the cellulose remaining predominant. In the b* coordinate (blue–yellow), all films presented a yellowish tone.

## 4. Conclusions

The study demonstrates that *Eucalyptus* spp. bark, subjected to different chemical and physical pretreatments, is a promising source to produce biodegradable films with varied chemical, thermal, mechanical, and optical properties. The alkaline pretreatment resulted in films with a greater mechanical strength and thermal stability, while the bleaching pretreatment promoted increased crystallinity and lower hydrophilicity. Conversely, the hydrothermal pretreatment influenced the reduction in optical transmittance and the modification of the surface morphology, generating films with a greater light barrier capacity. Each biomass pretreatment conferred specific characteristics to the films, making them interesting for different applications, such as biodegradable packaging and materials with improved optical properties. To improve performance and sustainability, further optimization of the pretreatment conditions—such as adjusting temperature, time, and reagent concentrations—could enhance film properties, including mechanical strength and thermal stability. Additionally, the incorporation of natural fibers or nanoparticles could increase the functionality of the films, providing further advantages in terms of resistance and performance. The development of multilayer films or active coatings could extend their use, not only as food packaging but also for agricultural purposes, such as biodegradable seed coatings or soil moisture control. Thus, this work contributes to the development of sustainable solutions by valorizing agro-industrial waste and promoting the use of biopolymers to replace conventional polymers, in line with the principles of the circular economy. These solutions could significantly reduce plastic waste in the food packaging and agricultural sectors, demonstrating a path toward a more sustainable future.

## Figures and Tables

**Figure 1 polymers-17-00105-f001:**
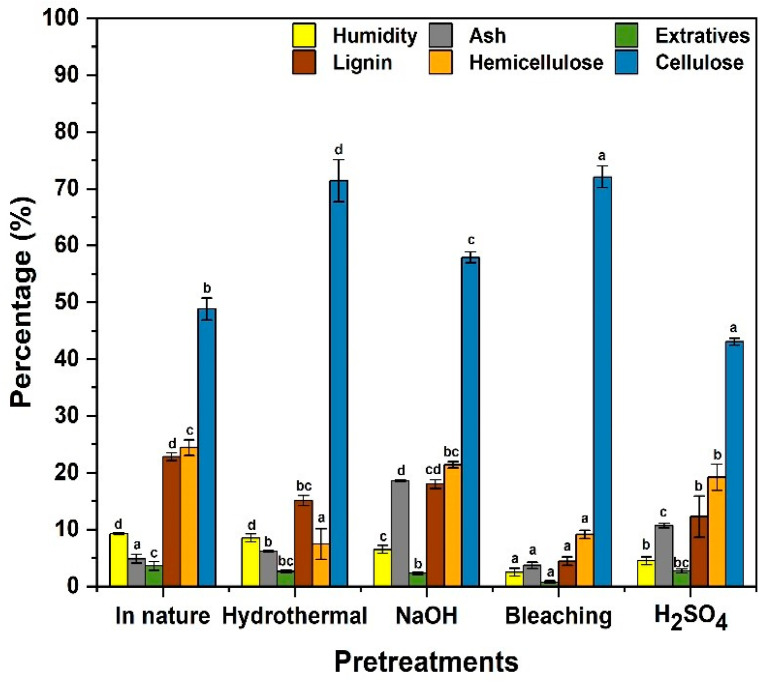
Mean values and standard deviations of the chemical constituents of the different analyzed samples of *Eucalyptus* spp. bark. The same letters above the values indicate non-significant statistical differences according to Fisher’s LSD test at a significance level of 5%.

**Figure 2 polymers-17-00105-f002:**
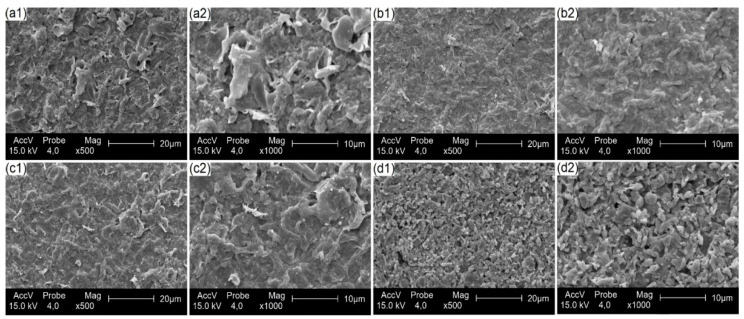
Micrographs at (**a1**,**b1**,**c1**,**d1**) 500× and (**a2**,**b2**,**c2**,**d2**) 1000× of the biomass films pretreated as follows: (**a1**,**a2**) hydrothermal, (**b1**,**b2**) NaOH, (**c1**,**c2**) bleaching, and (**d1**,**d2**) H_2_SO_4_.

**Figure 3 polymers-17-00105-f003:**
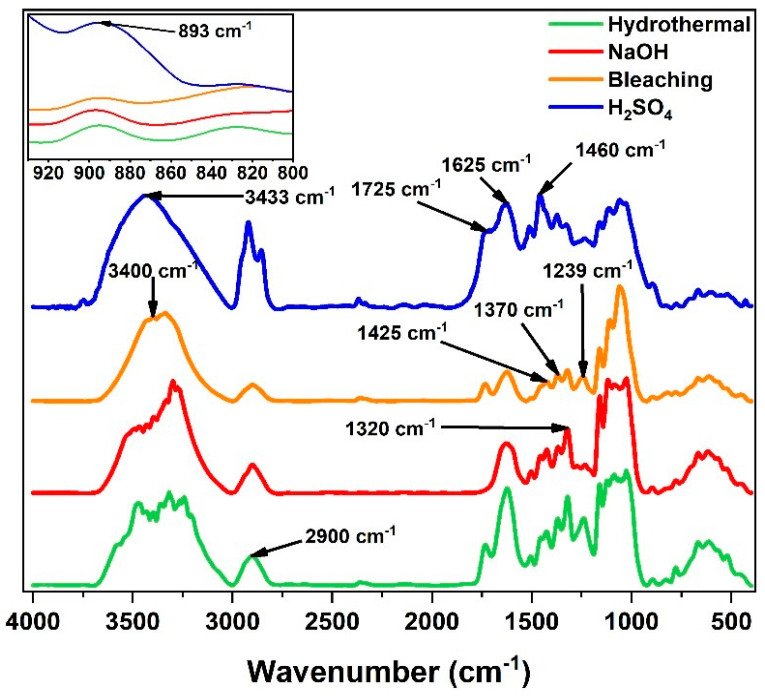
FTIR spectra of the films from the pretreated biomass.

**Figure 4 polymers-17-00105-f004:**
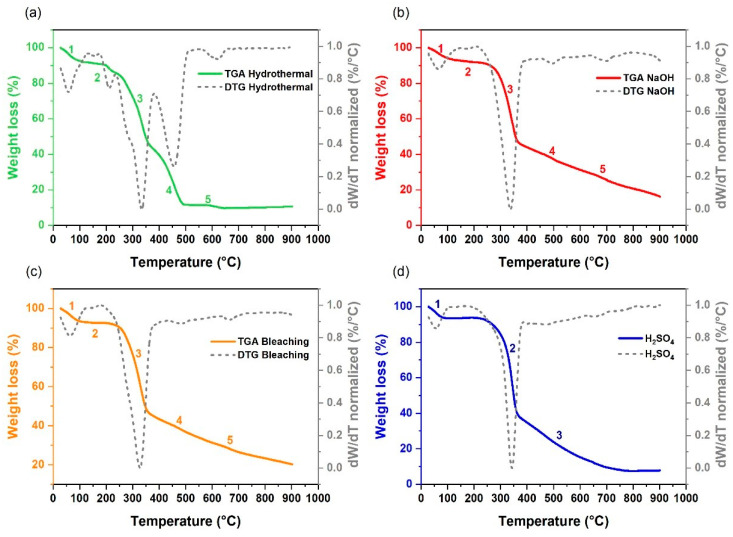
TGA analysis for films obtained from biomass subjected to different pretreatments: (**a**) hydrothermal, (**b**) NaOH, (**c**) NaClO, and (**d**) H_2_SO_4_.

**Figure 5 polymers-17-00105-f005:**
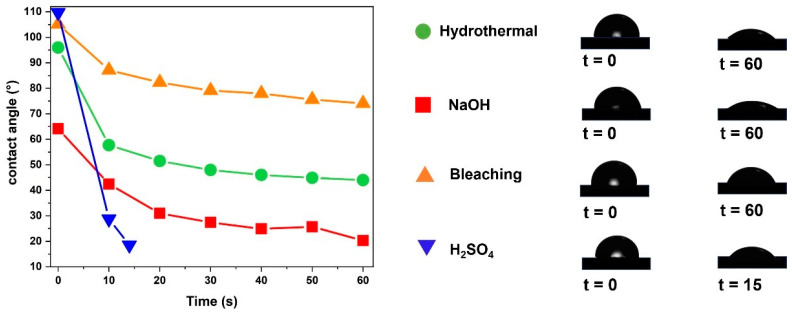
Wettability analysis of films from the pretreated biomass.

**Figure 6 polymers-17-00105-f006:**
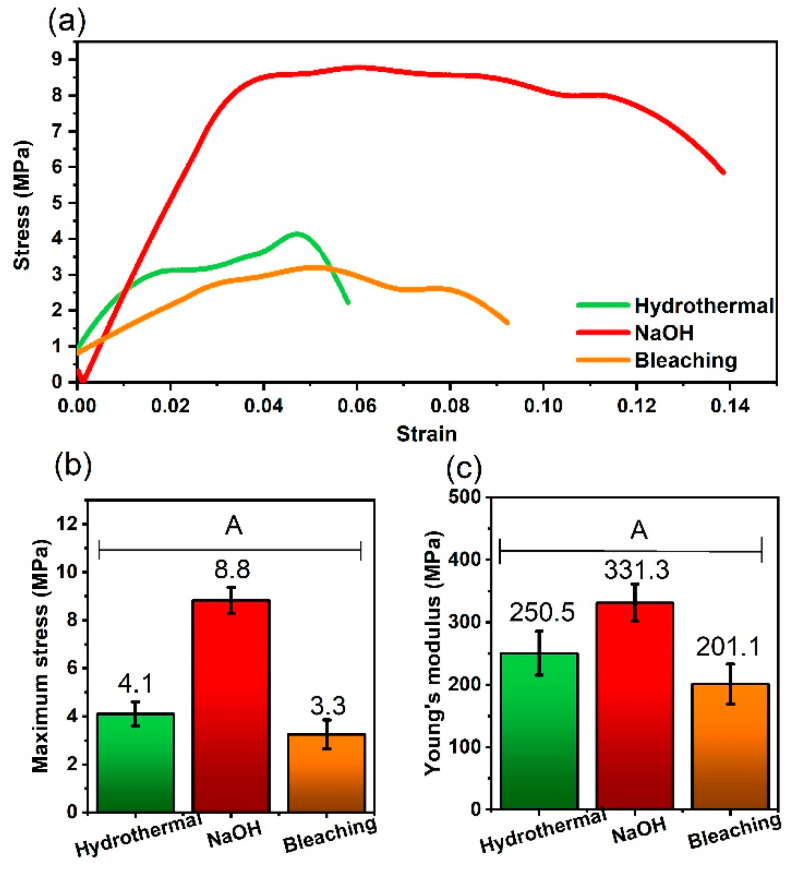
Mechanical properties in terms of (**a**) stress–strain curves, (**b**) maximum stress, and (**c**) Young’s modulus of the films produced from differently pretreated biomass. The same letters above the bar charts indicate non-significant statistical differences according to a Fisher’s LSD test at a significance level of 5%.

**Figure 7 polymers-17-00105-f007:**
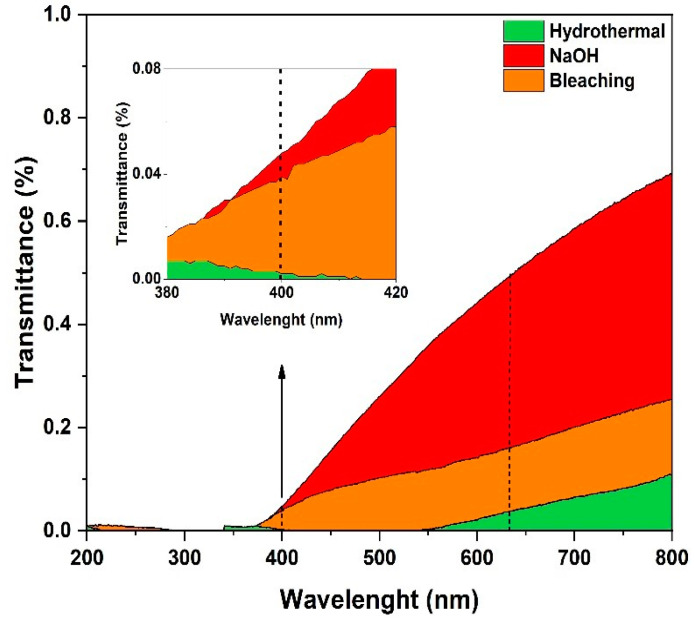
UV–Vis spectra of the films from the pretreated biomass.

**Figure 8 polymers-17-00105-f008:**
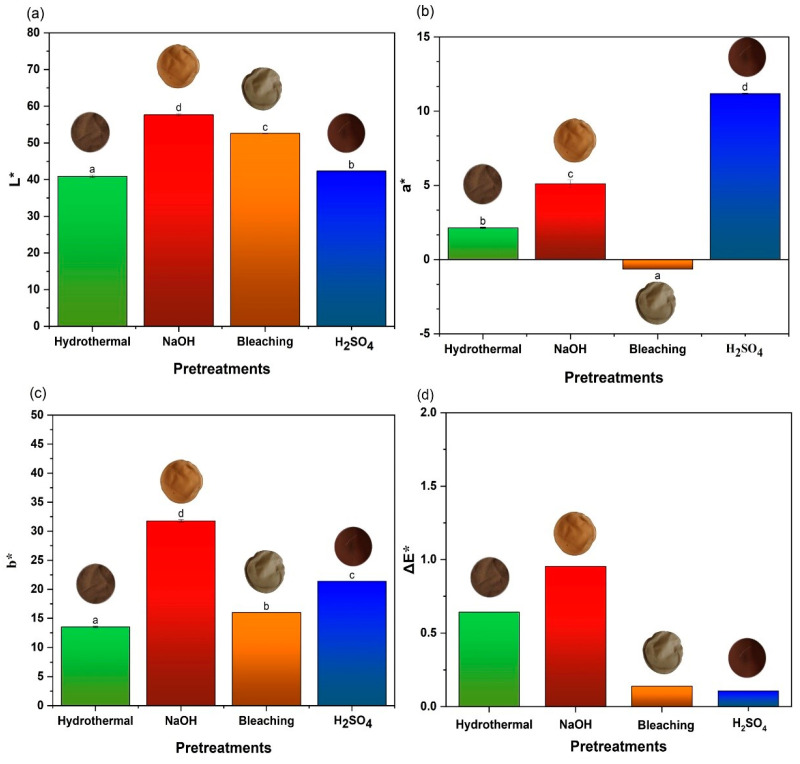
Colorimetry: (**a**) luminosity (L*), (**b**) green–red coordinate (a*), (**c**) blue–yellow coordinate (b*), (**d**) variation of all colors (ΔE*). The same letters above the bar charts indicate statistically insignificant differences according to a Fisher’s LSD test at a 5% significance level.

**Table 1 polymers-17-00105-t001:** FTIR spectral range of cellulose, hemicellulose, and lignin.

Spectral Range (cm^−1^)	Vibration	Functional Group	Chemical Compound	Reference
3500–3000	O–H stretching	Hydroxyls	Cellulose, hemicellulose, lignin	[54,55,57,58,59]
1870–1540	C=O stretching	Carbonyls	Hemicellulose	[55,57,59]
2900	C–H stretching	Aliphatic C–H	Cellulose, hemicellulose, lignin	[54,55,57,58,59]
1700–1400	C=C stretching	Aromatic ring (C=C)	Lignin	[55,57,59]
1500–1000	C–O–C stretching	C–O–C (glucose/ester)	Cellulose, hemicellulose, lignin	[54,55,57,58,59]
900–500	C–H stretching	C–H in aromatic rings	Lignin	[54,55,59]

**Table 2 polymers-17-00105-t002:** Tonset and Tpeak values obtained for the main stage of thermal decomposition of the films analyzed by TGA.

	Main Stage		
Films Obtained by Pretreatment	Tonset (°C)	Tpeak (°C)	Mass Loss (%)
Hydrothermal	302.38	351.64	41.14
NaOH	300.53	361.24	50.11
NaClO	281.70	350.48	50.44
H_2_SO_4_	311.65	357.42	64.54

**Table 3 polymers-17-00105-t003:** Transmittance values of the films from the pretreated biomass.

Films Obtained by Pretreatment	T (%) at 400 nm	T (%) at 633 nm
Hydrothermal	0.002	0.038
NaOH	0.048	0.492
Bleaching	0.039	0.160

## Data Availability

The original contributions presented in this study are included in the article. Further inquiries can be directed to the corresponding author.

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
