# Peer review of "Sustainable Films Derived from *Eucalyptus* spp. Bark: Improving Properties Through Chemical and Physical Pretreatments"

_polymers, 2025, doi:10.3390/polym17010105_

Round 1

Reviewer 1 Report

Comments and Suggestions for Authors

In this manuscript "Sustainable Development of Films from Eucalyptus spp. Bark " the authors reported about the development of sustainable films derived from Eucalyptus spp. Bark with mentioning about enhanced Properties.

However, I think the paper is still showing some weakness in various perspectives. Therefore, I recommend the authors to consider the following comments to address before publication.

The title is so simple, need to revise and can find one suggestion as " Sustainable Biodegradable Films Derived from Eucalyptus spp. Bark: Enhancing Properties via Chemical and Physical Pretreatments"

The abstract should be rewritten well by incorporating a sentence about the challenges of waste valorization or the need for sustainable materials and some other key findings.

Need to provide useful write-up including more recent studies, Focus on the Problem, Emphasize Novelty, context of application and research gaps in the literature that would be helpful for this research. And avoid Repetitive Content.

Need to include more information about the analysis, such as how SEM images were take or processed, how FTIR spectra were analyzed/interpreted, and how thermal stability was quantified.

Discuss how the findings of this were align or differ from prior studies.

For example, "The observed mechanical strength of 8.8 MPa for alkaline-treated films is consistent with similar findings by [Author, Year], who reported a range of ..........for similar materials." and so on.

The discussion based on the results looks like without deeply analyzing the implications or mechanisms behind the findings. So need to explain properly, why certain trends were observed.

Likewise, There is a lack of in-depth comparison with similar studies. For instance, how do the mechanical properties or other concern  properties of these films compare to other biodegradable films in the literature?

Need to add error bars to all results.

Need to add AFM results.

The discussion does not clearly state how the findings of this study contribute to advancing this research field or what gaps still need to be addressed, so need to incorporate accordingly.

Please ensure all references are formatted accordingly, and check all errors in term of grammatical or typo errors.

Comments on the Quality of English Language

Need to improve.

Author Response

Reviewer #1

Comments and Suggestions for Authors

In this manuscript "Sustainable Development of Films from Eucalyptus spp. Bark " the authors reported about the development of sustainable films derived from Eucalyptus spp. Bark with mentioning about enhanced Properties. However, I think the paper is still showing some weakness in various perspectives. Therefore, I recommend the authors to consider the following comments to address before publication.

  1. The title is so simple, need to revise and can find one suggestion as "Sustainable Biodegradable Films Derived from Eucalyptus spp. Bark: Enhancing Properties via Chemical and Physical Pretreatments"

Dear Reviewer, we appreciate your constructive suggestion regarding the title of the manuscript. We agree that a more detailed title could better reflect the focus and scope of the work. Based on your recommendation, we have revised the title to more clearly specify the aspects investigated. The new proposed title is:

"Sustainable Films Derived from Eucalyptus spp. Bark: Improving Properties through Chemical and Physical Pretreatments."

  1. The abstract should be rewritten well by incorporating a sentence about the challenges of waste valorization or the need for sustainable materials and some other key findings.

Dear Reviewer, We appreciate your comments on how to improve the summary. We have incorporated a sentence into the text that addresses the challenges of waste recovery and the need for sustainable materials:

"The valorization of agro-industrial residues and the increasing demand for sustainable materials represent significant challenges for the development of environmentally responsible solutions."

Furthermore, we highlight the contribution of the study to sustainable development and the circular economy, as well as the main findings, as suggested:

"This study contributes to sustainable development by valorizing agro-industrial residues and advancing the circular economy, thereby providing innovative solutions to environmental and industrial challenges. The findings indicate that the films exhibit specific properties that render them interesting for diverse applications, including sustainable packaging."

  1. Need to provide useful write-up including more recent studies, Focus on the Problem, Emphasize Novelty, context of application and research gaps in the literature that would be helpful for this research. And avoid Repetitive Content.

We appreciate the constructive comments on the introduction. We have made the suggested changes to improve the writing. We have added more recent and relevant studies to contextualize the topic, emphasized the problem and the novelty of the research, and provided a clear context for application. In addition, we have identified specific gaps in the literature that our research aims to address. We have also revised the text to avoid repetition and ensure greater objectivity and clarity.

  1. Need to include more information about the analysis, such as how SEM images were take or processed, how FTIR spectra were analyzed/interpreted, and how thermal stability was quantified.

We appreciate the observation about the need for greater detailing of the methodological techniques. We have revised the corresponding sections and included more specific information, such as the procedure for obtaining and processing SEM images, the approach used for analyzing and interpreting FTIR spectra, and the methodology used to quantify thermal stability.

  1. Discuss how the findings of this were align or differ from prior studies.

For example, "The observed mechanical strength of 8.8 MPa for alkaline-treated films is consistent with similar findings by [Author, Year], who reported a range of ..........for similar materials." and so on.

The discussion based on the results looks like without deeply analyzing the implications or mechanisms behind the findings. So need to explain properly, why certain trends were observed.

Likewise, There is a lack of in-depth comparison with similar studies. For instance, how do the mechanical properties or other concern  properties of these films compare to other biodegradable films in the literature?

We appreciate your valuable comments and fully agree on the importance of deepening the discussion, including more detailed comparisons with previous studies and clear explanations for the observed trends. In response to your feedback, we have revised the discussion section 3.6 Tensile Strength Analysis.

  1. Need to add error bars to all results.

Thank you for your valuable feedback on the tensile strength graphs. Please note that the graphs have been reviewed and corrected as indicated. Additionally, we have added standard deviation bars to provide better visualization and interpretation of the data.

  1. Need to add AFM results.

We appreciate your valuable suggestion regarding the inclusion of atomic force microscopy (AFM) results to complement the study. We recognize that this analysis would be relevant to further characterize the films. However, due to the limited time to respond to the reviews and the unavailability of resources, it was not possible to perform this technique at this time.

To partially meet your request, we highlight that the work presents scanning electron microscopy (SEM) results, which allow us to observe and discuss the surface morphology of the treated films. We believe that these data contribute to the understanding of the characteristics investigated.

  1. The discussion does not clearly state how the findings of this study contribute to advancing this research field or what gaps still need to be addressed, so need to incorporate accordingly.

We are grateful for the comment that allowed us to reflect on the contribution of the study to the advancement of the research field and the gaps that still exist. We have made adjustments to the text to incorporate this information in a clear and detailed manner.

We have added a discussion on how the findings of this study demonstrate the potential of biodegradable films derived from biomass to replace synthetic polymers in applications that require specific mechanical, thermal, and optical properties, directly contributing to the development of more sustainable materials. We also highlight that the approach used in this work, without additives such as plasticizers or stabilizers, expands the relevance in the literature, while emphasizing the need to further investigate the impact of variables such as temperature, reagent concentration, and the combination of different pretreatments to improve film performance.

  1. Please ensure all references are formatted accordingly, and check all errors in term of grammatical or typo errors.

We appreciate your attention to detail, which contributes to the quality of the work. We performed a thorough review to ensure that all references were properly formatted according to the journal's guidelines. In addition, we carefully checked the manuscript to correct any grammatical or typographical errors that may have been present.

Reviewer 2 Report

Comments and Suggestions for Authors

Overall the paper shows some of novelty elements and good for advancement in the field. However, many rooms for improvement can be imposed as such clearer focus will make the discussion easier to follow and connect the findings to practical applications. Hence, the improvement can be done:

   - Clarify how the removal of lignin and hemicellulose or exposure of hydroxyl groups influences mechanical, thermal, and optical properties.  

   - Provide a simple explanation of the molecular-level changes caused by different pretreatments.

   - Discuss the advantages and disadvantages of each pretreatment method such as NaOH, bleaching, hydrothermal.  

   - Highlight trade-offs between mechanical strength, optical clarity, and environmental impact.

   - Link the TGA results to real-world uses, such as packaging for hot or cold environments.  

   - Mention which films are suitable for high-temperature applications and why.

   - Explain how the morphology and crystallinity of the films affect their light-scattering and absorption properties.  

   - Compare these findings with common materials used for light barriers.

   - Compare the mechanical strength of the films to existing biodegradable or synthetic materials.  

   - Discuss practical uses where the NaOH-treated films might perform better due to higher strength.

   - Address challenges like brittleness in acid-treated films or the high chemical usage in bleaching.  

   - Suggest ways to improve these processes for better performance and sustainability.

   - Expand on how these films can replace conventional materials in industries like food packaging or agriculture.  

   - Discuss how the findings contribute to the circular economy and sustainability goals.

Comments on the Quality of English Language

Need to proofread the article by native English speaker to make more coherent in the discussion

Author Response

Comments and Suggestions for Authors

Overall the paper shows some of novelty elements and good for advancement in the field. However, many rooms for improvement can be imposed as such clearer focus will make the discussion easier to follow and connect the findings to practical applications. Hence, the improvement can be done:

  1. Clarify how the removal of lignin and hemicellulose or exposure of hydroxyl groups influences mechanical, thermal, and optical properties. 

We appreciate your suggestion for clarification on the impact of lignin and hemicellulose removal, as well as the exposure of hydroxyl groups, on the mechanical, thermal and optical properties of the films. In response to your comment, we have provided the appropriate explanations in the relevant points of the manuscript. Below, we highlight the main clarifications included:

In section 3.6 Tensile strength analysis, we explain that lignin and hemicellulose removal favors direct interaction between cellulose fibers, increasing the formation of hydrogen bonds between exposed hydroxyl groups. This results in a more cohesive and less porous matrix, contributing to increased mechanical strength, as observed in films treated with NaOH.

In section 3.4 Thermogravimetric analysis, we detail how lignin and hemicellulose removal simplifies the structure of the material, increasing its thermal stability by reducing thermally unstable components. Furthermore, the exposure of hydroxyls contributes to greater structural cohesion, delaying the onset of thermal manipulation, as observed in the NaOH-treated films, which exhibited greater thermal stability.

In section 3.7 Transmittance Analysis, we discussed that the removal of lignin, rich in chromophores responsible for light capture, improves the transparency of the films, while the exposure of hydroxyls influences the molecular organization and reduces light refraction. In the films treated with NaOH and bleaching, we observed greater optical clarity due to the reduction of lignin and better fiber alignment.

  1. Provide a simple explanation of the molecular-level changes caused by different pretreatments.

We are grateful for the observation and the opportunity to clarify the changes at the molecular level caused by the different pretreatments. For the pretreatment with NaOH, we consider that the alkaline action probably caused the rupture of the C=C and C=O bonds of the aromatic rings present in the lignin structure, contributing to the structural modification of the material. This explanation was added to item 3.3 FTIR Analysis, where we emphasize that such changes “may have caused the rupture of the bonds”, aligning the results of the analyses with the expected chemical transformations.

  1. Discuss the advantages and disadvantages of each pretreatment method such as NaOH, bleaching, hydrothermal.  

We appreciate the suggestion to discuss the advantages and disadvantages of pretreatment methods such as NaOH, bleaching, and hydrothermal. We have made the requested changes to the Introduction. The section now covers in detail the benefits and challenges of each method, considering aspects such as efficiency, environmental impact, and operating costs, as detailed in the highlighted paragraph below:

“Therefore, chemical pretreatments are essential in this process, as they facilitate the separation and purification of the main components of the biomass [22]. Methods such as acid and alkaline hydrolysis play a crucial role, as acid hydrolysis degrades hemicellulose and part of the lignin, while alkaline hydrolysis removes lignin more efficiently [23–25]. These treatments, along with other chemical and physical processes, such as bleaching, aim to improve film-forming properties such as mechanical strength, transparency, and crystallinity. Hydrothermal pretreatment, which uses only water and heat, stands out as a more sustainable option, as it eliminates the need for chemical agents in film formation [26]. Among the advantages of these processes are their efficiency in cellulose purification and improvement of film properties, making them interesting for various industrial applications [15]. Furthermore, these methods offer greater flexibility, enabling adaptation of the material's characteristics according to the specific needs of each application. However, challenges include environmental impacts associated with the use of strong acids and chlorinated agents, which generate potentially harmful residues, requiring proper treatment to minimize contamination. Another challenge is the high energy and time consumption, especially in thermal and chemical processes, which can increase operational costs.”

  1. Highlight trade-offs between mechanical strength, optical clarity, and environmental impact.

We appreciate your valuable suggestion to address the trade-offs between mechanical strength, optical clarity, and environmental impact. We have incorporated these discussions into the manuscript, highlighting the following points:

Mechanical Strength vs. Optical Clarity:

As discussed in sections 3.6 Tensile Strength Analysis and 3.7 Transmittance Analysis, we observed that NaOH-treated films showed higher mechanical strength (8.8 MPa) due to the partial retention of lignin and hemicellulose, which reinforces the material matrix. However, this composition results in lower optical clarity compared to bleached films, which have lower lignin content but exhibit higher transparency due to the removal of chromophores.

Optical Clarity vs. Environmental Impact:

Bleached films showed good optical clarity, but this process uses oxidizing agents (NaClO), which can generate chemical waste with greater environmental impact. On the other hand, hydrothermal treatment, despite producing less transparent films, is a more environmentally friendly alternative because it does not use aggressive chemicals, demonstrating a more sustainable approach.

Mechanical Resistance vs. Environmental Impact:

Treatment with NaOH, which promotes greater mechanical resistance, involves the use of strong bases, which require adequate waste treatment. In contrast, hydrothermal treatment, despite resulting in lower resistance (4.1 MPa), has a lower environmental impact because it uses only water and heat.

These trade-offs were highlighted in the text to emphasize that the choice of treatment method should be guided by the specific priorities of the final application, considering the balance between technical performance and environmental sustainability.

  1. Link the TGA results to real-world uses, such as packaging for hot or cold environments.  

We are grateful for the suggestion to link the TGA results to real-world applications, such as packaging for hot or cold environments. Based on the guidance, we expanded the discussion of the obtained data by incorporating an analysis of the suitability of the films for different practical conditions.

  1. Mention which films are suitable for high-temperature applications and why.

We are grateful for the observation and the opportunity to improve the manuscript. As requested, we have included a detailed analysis in the discussion, clarifying which films are more suitable for high-temperature applications and the underlying reasons.

The films treated with NaOH demonstrated greater thermal stability, concentrating the main decomposition stage at higher temperature ranges, as observed in the TGA results. This characteristic is attributed to the effective removal of amorphous components, such as hemicellulose, during the alkaline treatment, as well as to the exposure of hydroxyl groups, which increase the structural cohesion of the polymer matrix. This greater stability makes these films more suitable for applications that require resistance to high temperatures, such as packaging for processed foods, reheated or subjected to thermal processes during transportation and storage.

We hope that the additional information meets the expectations and provides greater clarity on the practical applicability of the studied materials.

  1. Explain how the morphology and crystallinity of the films affect their light-scattering and absorption properties.  

We are grateful for the valuable suggestion to address the relationship between the morphology, crystallinity of films and their light scattering and absorption properties. This issue was discussed mainly in items 3.2 Morphological Analysis and 3.7 Transmittance Analysis, as described below:

In item 3.2 Morphological Analysis, we highlighted that irregular surfaces, such as those observed in films obtained by hydrothermal treatment, increase light scattering due to the presence of multiple points of reflection and refraction. In contrast, films with more compact surfaces, such as those treated with NaOH, presented less light scattering, due to their lower roughness and greater morphological uniformity.

In item 3.7 Transmittance Analysis, we explained that the crystallinity of films directly affects light absorption, since more crystalline structures present greater density and molecular ordering, which can intensify light absorption and reduce transmittance. This is evidenced by the low transmittance values ​​of the hydrothermal films (0.002% at 400 nm), associated with greater crystallinity, while the films treated with NaOH showed greater transmittance due to lower crystallinity and a more uniform surface.

  1. Compare these findings with common materials used for light barriers.

We appreciate the suggestion to compare the light scattering and absorption properties of the films with materials commonly used as light barriers. This comparison was added to the manuscript in item 3.7 Transmittance Analysis. We highlight that the films developed in this study, especially those obtained by hydrothermal treatment, showed excellent light blocking capacity (transmittance of 0.002% at 400 nm), comparable to opaque synthetic polymers and metallized films. However, unlike these conventional materials, which often rely on additives or metal coating processes, biomass films offer the advantage of being renewable and biodegradable, aligning technical performance with environmental sustainability.

  1. Compare the mechanical strength of the films to existing biodegradable or synthetic materials.  

We appreciate your important suggestion regarding the comparison of the mechanical strength of films with existing biodegradable or synthetic materials. In response to your recommendation, we have included this detailed comparative analysis in section 3.6 Tensile Strength Analysis.  In this section, in addition to presenting the results of the tensile strength analyses of the films produced, we have provided a comprehensive discussion comparing our data with the values ​​reported for other biodegradable materials (such as films based on starches, chitosans and modified cellulose) and some synthetic polymers widely used in industry.

  1. Discuss practical uses where the NaOH-treated films might perform better due to higher strength.

Thank you for your suggestion. In response to your recommendation, we have added the following paragraph to address practical applications:

"NaOH-treated films, due to their increased mechanical strength, are particularly suitable for applications where a robust and durable material is required. Examples include packaging for the transport of heavy or sharp products, such as metal tools and utensils, and industrial applications where materials are subject to significant mechanical stress. In addition, their high strength makes them ideal for use in protective films that require durability under adverse conditions, such as in logistics processes or prolonged storage."

  1. Address challenges like brittleness in acid-treated films or the high chemical usage in bleaching.  

We appreciate the pertinent observation about the challenges related to the brittleness of acid-treated films and the high use of chemicals in the bleaching process. We acknowledge that the treatment with H₂SO₄ resulted in more brittle films due to the degradation of the cellulose chains, as described in item 3.6 Tensile Strength Analysis. This aspect was duly mentioned in the text, highlighting the limitations for applications that require greater mechanical strength.

Regarding bleaching, we emphasize that, although the use of NaClO promoted the effective removal of lignin and hemicellulose, we recognize the environmental concerns associated with the use of chemical oxidizing agents. We have included a reflection on these limitations in the revised paragraph, reinforcing the need for future optimization of chemical treatments to balance technical performance and environmental impact.

  1. Suggest ways to improve these processes for better performance and sustainability.

We appreciate the suggestion to suggest improvements in the processes to achieve better performance and sustainability. We have incorporated proposals to improve pretreatments into the article, as detailed in the highlighted paragraph below:

“To improve performance and sustainability, further optimization of pretreatment conditions—such as adjusting temperature, time, and reagent concentration—could enhance film properties, including mechanical strength and thermal stability. Additionally, the incorporation of natural fibers or nanoparticles could increase the functionality of the films, providing further advantages in terms of resistance and performance. The development of multilayer films or active coatings could extend their use, not only as food packaging but also for agricultural purposes, such as biodegradable seed coatings or soil moisture control. Thus, this work contributes to the development of sustainable solutions by valorizing agro-industrial waste and promoting the use of biopolymers to replace conventional polymers, in line with the principles of the circular economy. These solutions could significantly reduce plastic waste in food packaging and agriculture, demonstrating a path toward a more sustainable future.”

  1. Expand on how these films can replace conventional materials in industries like food packaging or agriculture.  

We appreciate the suggestion to expand the discussion on how films can replace conventional materials in industries such as food packaging or agriculture. We have incorporated the suggested changes into the article, as detailed in the highlighted paragraph below:

“Films developed from the bark of Eucalyptus spp. have specific properties that make them promising alternatives to replace conventional polymers in several industries. In the food packaging industry, these biodegradable films offer advantages such as light barrier, mechanical resistance and thermal properties that can be adjusted to store perishable foods or package light-sensitive products. In addition, due to their biodegradability, these materials can significantly reduce the environmental impact associated with the disposal of conventional plastics. In the agricultural sector, the films can be applied as biodegradable seed coatings or as films for soil moisture control, contributing to the reduction of the use of non-renewable plastics and promoting more sustainable agricultural practices. The versatility of the adjustable properties of these materials expands their applicability, consolidating their potential as environmentally friendly solutions to replace petroleum-derived polymers.”

  1. Discuss how the findings contribute to the circular economy and sustainability goals.

We appreciate the suggestion to discuss how the findings contribute to the circular economy and sustainability goals. We have already incorporated this discussion into the paper, as detailed in the highlighted paragraph below:

“This study significantly contributes to the valorization of agro-industrial waste, such as Eucalyptus spp. bark, through the implementation of sustainable practices aligned with the principles of the circular economy. The transformation of this material into biodegradable films through chemical and physical pretreatments demonstrates an innovative and efficient approach that combines advanced technologies with environmental responsibility.

Therefore, this study directly contributes to the Sustainable Development Goals (SDGs), especially SDG 12 (Responsible Consumption and Production), by promoting circular economy practices and waste reduction, using Eucalyptus spp. bark to produce renewable materials. Furthermore, by applying technologies that enhance lignocellulosic structure, the work supports SDG 9 (Industry, Innovation, and Infrastructure), developing biodegradable alternatives to replace conventional polymers. The study also supports SDG 13 (Climate Action) by reducing dependence on non-renewable materials and mitigating environmental impacts, such as carbon emissions and plastic pollution [27]. Thus, this work demonstrates how scientific and technological advancements can align technical efficiency with global sustainability goals, contributing to a more responsible and ecological future.”

Reviewer 3 Report

Comments and Suggestions for Authors

In my opinion, the manuscript can fit the scope of the journal. Biobased films from lignocellulosic biomasses are a valuable topic for sustainable development. Moreover, the English is clear, and self-citations are fine. The manuscript is well written; I commented on the introduction about the references. The experimental part is clear. I have only a few minor suggestions. The results are well discussed; I have only one comment in the FTIR part and one in the TGA part. Please check the use of subscripts, as in the case of H2SO4.

Line 19: the CI is reducing or increasing? Please clarify in the text.

Line 65: “Therefore, chemical pretreatments are essential in this process, as they facilitate the separation and purification of the main components of the biomass”. I recommend adding literature to support the sentence. As a practical recent example of chemical pretreatment for component extraction, I suggest:

Polymers 2023, 15(17), 3591; https://doi.org/10.3390/polym15173591

Line 78: The authors state that “there are few reports”. I suggest adding at the end of the sentence the references to these few reports. It will help the reader.

Line 88: I suggest adding “… Pelotas-RS, Brazil.”. It will help the international audience.

Line 89: H2SO4 subscripts. Please check the whole manuscript for all the subscripts.

Line 100: I suggest removing “with sulfuric 100 acid, sodium hydroxide, bleaching and hydrothermal”. The authors report below all the treatments.

Line 339: I suggest adding a table or writing a paragraph reporting the main band associated specifically with cellulose, hemicellulose and lignin. It will help the reader to understand the discussion of the spectra.

Line 392: I suggest adding a table with the measurable Tonset and Tpeak values.

Line 657: I suggest changing the words “making them suitable” to “making them interesting”. The reason is that further analyses are necessary before establishing suitable applications.

Author Response

Comments and Suggestions for Authors

In my opinion, the manuscript can fit the scope of the journal. Biobased films from lignocellulosic biomasses are a valuable topic for sustainable development. Moreover, the English is clear, and self-citations are fine. The manuscript is well written; I commented on the introduction about the references. The experimental part is clear. I have only a few minor suggestions. The results are well discussed; I have only one comment in the FTIR part and one in the TGA part.

  1. Please check the use of subscripts, as in the case of H2SO4.

We appreciate your attention to detail, which helps improve the quality of the work. We have carefully reviewed the manuscript and corrected the use of subscripts.

  1. Line 19: the CI is reducing or increasing? Please clarify in the text.

Thank you for your observation about the crystallinity index. In fact, we have identified that the term was misspelled. For clarity, we have replaced the term “increase in crystallinity” with “highest crystallinity index”.

  1. Line 65: “Therefore, chemical pretreatments are essential in this process, as they facilitate the separation and purification of the main components of the biomass”. I recommend adding literature to support the sentence. As a practical recent example of chemical pretreatment for component extraction, I suggest: Polymers 2023, 15(17), 3591; https://doi.org/10.3390/polym15173591

We are grateful for the valuable observation and for indicating the reference. As suggested, we included the citation of the article https://doi.org/10.3390/polym15173591

in the text to support the statement about the importance of chemical pretreatments in the separation and purification of the main components of biomass. This addition contributes to reinforce the theoretical basis of the work.

  1. Line 78: The authors state that “there are few reports”. I suggest adding at the end of the sentence the references to these few reports. It will help the reader.

We appreciate your valuable suggestion. However, since this is a new study, there were no additional references to include. To avoid misunderstandings, we have modified the wording of the paragraph as follows:

“Although each pretreatment technique has different yields, advantages, and limitations, this study aims to explore the potential of Eucalyptus spp. bark for film production, applying chemical and physical pretreatments to optimize the extraction of lignocellulosic biomass constituents and improve the properties of the films. The uniqueness of this research lies in the combination of chemical and physical techniques without the addition of additives, such as crosslinkers, plasticizers, or stabilizers, for film formation, expanding its relevance in the literature.”

  1. Line 88: I suggest adding “… Pelotas-RS, Brazil.”. It will help the international audience.

Thank you for your valuable suggestion. We have made the recommended change and added “Pelotas-RS, Brazil” to better guide international audiences.

  1. Line 89: H2SO Please check the whole manuscript for all the subscripts.

Thank you for noting the subscripts in the manuscript. We have done a thorough review of the text to ensure that all subscripts are properly formatted.

  1. Line 100: I suggest removing “with sulfuric 100 acid, sodium hydroxide, bleaching and hydrothermal”. The authors report below all the treatments.

Thank you for your valuable suggestion. We have made the recommended change and removed the phrase “with 100% sulfuric acid, sodium hydroxide, bleaching and hydrothermal”.

  1. Line 339: I suggest adding a table or writing a paragraph reporting the main band associated specifically with cellulose, hemicellulose and lignin. It will help the reader to understand the discussion of the spectra.

We appreciate the suggestion to include specific information about the bands associated with cellulose, hemicellulose, and lignin. Following your recommendation, we have added a brief commentary in the FTIR analysis section and included a table specifically reporting the major bands corresponding to these components.

  1. Line 392: I suggest adding a table with the measurable Tonset and Tpeak values.

We appreciate your suggestion. In response to your comment, we have added to the manuscript a Table containing the measurable values ​​of Tonset and Tpeak, corresponding to the main stage of the TGA results. We hope that this inclusion will contribute to a better understanding of the data presented.

  1. Line 657: I suggest changing the words “making them suitable” to “making them interesting”. The reason is that further analyses are necessary before establishing suitable applications.

We appreciate the suggestion to replace the phrase “making them suitable” with “making them interesting.” We agree that additional analysis is needed before establishing suitable applications, so we have made the change as recommended.

Round 2

Reviewer 2 Report

Comments and Suggestions for Authors

Well done

Comments on the Quality of English Language

Improve and need to be proofread

Reviewer 3 Report

Comments and Suggestions for Authors

The authors replied to all my comments and suggestions. In my opinion, the manuscript can be accepted.